

# Identification of core sub-team on scientific collaboration networks with Shapley method

Lixin Zhou[1,2], Chen Liu[1,2] and Xue Song[2]

[1] School of Intelligent Emergency Management, University of Shanghai for Science and Technology, Shanghai, China
[2] Business School, University of Shanghai for Science and Technology, Shanghai, China

## ABSTRACT

Identifying the core sub-teams that drive productivity in scientific collaboration networks is essential for research evaluation and team management. However, existing methods typically rank individual researchers by bibliometric impact or select structurally cohesive clusters, but rarely account for both collaboration patterns and joint scientific output. To address this limitation, we propose a novel two-dimensional framework that integrates network topology with research performance to identify core sub-teams. Specifically, we measure each sub-team's marginal structural contribution using the Shapley value and quantify its collective impact using a sub-team H-index. To efficiently identify high-contributing sub-teams, we employ the Monte Carlo Tree Search algorithm, along with an approximation strategy to estimate Shapley values under computational constraints. We evaluate our method on 61 real-world scientific collaboration teams from Web of Science and Baidu Scholar data. Experimental results validate the effectiveness of our method in identifying core sub-teams, with the highest collaborative and citation impact. The proposed method offers a valuable analytical tool for research managers and funding agencies seeking to locate high-impact collaborative clusters, and it provides a generalizable framework for studies requiring the integration of structural and performance-based indicators in network analysis.

## INTRODUCTION

With the increasing complexity and uncertainty of scientific research, research collaboration has become a key approach for scholars to conduct research activities (*Essers, Grigoli & Pugacheva, 2022*). To better understand collaboration patterns, many studies have constructed scientific collaboration networks, where nodes represent researchers and edges signify collaborative relationships established through co-authorship of articles, books, or patents (*Judijanto, Suryadi & Tanjung, 2024*; *Chen, Zhang & Fu, 2019*) Within these networks, core sub-teams, formed by key researchers, play a crucial role in driving research activities and significantly influence the overall network (*Fortunato et al., 2018*; *HabibAgahi, Kermani & Maghsoudi, 2022*). Therefore, identifying those core sub-teams is of great theoretical and practical value for assessing research performance for building and

Corresponding author
Chen Liu, chenliu@usst.edu.cn

managing effective research teams (*McCambridge & Golder, 2024*; *Isfandyari-Moghaddam et al., 2023*; *Schäfermeier, Hirth & Hanika, 2023*).

Numerous scholars have investigated the evaluation and identification of influential sub-teams within scientific collaboration networks (*Gao et al., 2024*; *Kong et al., 2019*). Traditional methods such as the Delphi method, questionnaires, and expert brainstorming have been used to identify key nodes. However, with the explosive growth of digital information, these methods increasingly inadequate for addressing contemporary research demands (*Essers, Grigoli & Pugacheva, 2022*; *Judijanto, Suryadi & Tanjung, 2024*). Recent research on scientific teams typically leverages computational technology to collect scientific publications from various disciplines as empirical data. This approach constructs scientific collaboration networks from the perspective of cooperation relationships, analyzes structural characteristics of scientific collaboration networks across different fields, and explores the patterns of scientific collaboration among different scholars. Social network analysis, association rules, and hierarchical clustering are used to divide scientific collaboration networks into different communities and identify research teams within these networks (*Chen, Zhang & Fu, 2019*; *Chen et al., 2025*). From a research output perspective, existing studies select bibliometric indicators such as the team's overall publication volume, citation count, and H-index to assess team performance. In addition, from a network structure perspective, those methods including betweenness centrality, cohesive subgroup analysis, and K-core are chosen to evaluate the performance of research teams and extract leading teams from scientific collaboration networks (*HabibAgahi, Kermani & Maghsoudi, 2022*; *McCambridge & Golder, 2024*; *Isfandyari-Moghaddam et al., 2023*). Researchers have constructed scientific collaboration networks in various fields such as management science and information science, extracting leading research teams with notable identification effectiveness (*HabibAgahi, Kermani & Maghsoudi, 2022*; *Schäfermeier, Hirth & Hanika, 2023*; *Gao et al., 2024*).

Furthermore, key researchers are often regarded as the core and soul of research teams, which makes the identification of these pivotal individuals a focal area of interest among scholars (*Paulo & Brambilla, 2025*). Specifically, research based on social network analysis selects network node importance evaluation indicators to measure scholars' academic influence, coordination ability, and cohesion ability in research collaboration networks, thereby identifying leaders and key personnel within research teams (*Camur & Vogiatzis, 2024*). To determine which nodes have the greatest impact in social networks, *Lazebnik, Beck & Shami (2023)* referred to solutions for the benefit distribution problem in team collaboration from game theory. They proposed calculating the Shapley value for each node to better identify key nodes and discover influencers in the network. Shapley value analysis is a classical benefit distribution method for solving cooperative N-Person game problems (*Shapley, 1953*), with a core idea that embodies the contribution of each component to the collective benefit. Currently, applications of the Shapley value are mostly concentrated in fields such as game theory, interpretability in machine learning, economic pricing, and social network analysis. *Lipovetsky (2021)* demonstrated that Shapley value analysis can be used to estimate the relative usefulness of regression coefficients and predictors in a model, facilitating the interpretation of model results and aiding

predictions. *Tian et al. (2022)* also clarified that Shapley value analysis can be used in economics to determine data boundaries and pricing, maximizing data utilization while protecting data privacy.

However, most existing methods suffer from two key limitations. First, they often focus on a single dimension—either bibliometric indicators (such as publications, citations, h-index) or structural metrics (such as group size, k-core, cohesive subgroups)—and often target individuals, thereby neglecting the collaborative relationships. Second, most studies target individual team members, overlooking the influence of collaborative relationships among key nodes, which results in a less accurate identification of core sub-teams at the network level.

To address the aforementioned issues, this article proposes a novel method for identifying core sub-teams based on the dynamics of scientific collaboration networks. Our method integrates both research impact metrics and network structural properties, using Shapley value and sub-network h-index as joint indicators. The Shapley value captures each sub-team's marginal contribution to the entire network, considering its interactions with other sub-teams. By iteratively removing nodes from the network and analyzing the resulting changes in structure and collaboration patterns, we estimate each sub-team's relative importance. To address the high computational complexity of this process, we introduce the Monte Carlo Tree Search (MCTS) algorithm, along with an approximation strategy to estimate Shapley values under computational constraints.

In summary, the contributions of this study are as follows:

- This article introduces a novel approach that integrates both bibliometric and network structural dimensions to identify core sub-teams in scientific collaboration networks. Unlike traditional methods that focus on a single dimensions, our method provides a comprehensive analysis that captures dynamic interactions and contributions within scientific teams.
- The study employs Shapley value analysis to evaluate the importance of team, and employ the MCTS algorithm, along with an approximation strategy to reduce the computational cost of Shapley value estimation.
- To evaluate the effectiveness of the proposed approach, we apply it to 61 real-world research teams derived from Web of Science and Baidu Scholar. Experimental results show our method outperforms traditional methods in identifying teams with the highest collaborative and citation impact.

## RELATED WORKS

### Identification subteams of scientific collaboration

Scientific collaboration networks are complex structures formed by the cooperative efforts of scientists and researchers. Identifying core teams within these networks has been approached from various perspectives. Traditional methods include rule-based and clustering algorithms, often focus on metrics like co-authorship and citation networks to identify key researchers and cohesive groups. These methods employ measures like centrality and community detection often visualizing results to enhance the reliability of

sub-team identification through human judgment (*McCambridge & Golder, 2024*; *Isfandyari-Moghaddam et al., 2023*).

Cluster analysis has been a vital method in identifying significant research teams within collaboration networks. For instance, *Ghafouri et al. (2012)* used cluster analysis to map the co-authorship network of Iranian emergency medicine, demonstrating its critical importance in delineating key collaborative groups and enhancing the visualization of complex research networks. Similarly, *Ji et al. (2022)* proposed methods for clustering authors based on their research interests. Their findings show that most researchers continue to collaborate with the same cluster of people over many years. Such methods have been instrumental in understanding the structure and dynamics of scientific collaboration.

The identification of research teams based on social network analysis (SNA) combines human cognitive abilities with computational techniques, enhancing both the credibility and accuracy of the results. *Gao et al. (2024)* illustrated the potential of identifying key researchers and collaboration patterns through co-authorship network analysis in health research. *Li et al. (2016)* used SNA to uncover significant collaboration patterns and nodes in global scientific collaboration, while *Isfandyari-Moghaddam et al. (2023)* emphasized the practicality of SNA and data mining techniques in identifying key groups and influential researchers at the global level. Overall, SNA provides comprehensive insights into research collaboration and connections but also effectively identifies key nodes and influencers through sophisticated metrics such as betweenness centrality. These methods visualize complex relationships to improve the accuracy and reliability of research team identification processes (*Isfandyari-Moghaddam et al., 2023*).

However, existing methods often overlook the intricate contributions of individual nodes when combined with others. To address this limitation, this article utilized the Shapley value, rooted in cooperative game theory, to assess the collective importance of interconnected researchers and quantify their individual contributions within the overall network.

## Researcher influence ranking

Identifying key researchers within academic teams has become a focal point for scholars, particularly through the use of social network analysis to evaluate important nodes within networks. This approach measures academic influence, collaboration capacity, and cohesion within scientific networks, highlighting leaders and key contributors (*Schäfermeier, Hirth & Hanika, 2023*; *Gao et al., 2024*).

Various centrality measures have been employed to rank researchers within scientific collaboration networks, each offering unique insights into the structure and dynamics of these networks. For instance, betweenness centrality has been widely used to evaluate researchers based on their intermediary roles—identifying individuals who serve as bridges between otherwise disconnected groups. This method has been shown to effectively highlight influential nodes in scientific collaborations (*Isfandyari-Moghaddam et al., 2023*). Another prominent approach is the application of game-theoretic methods, particularly the use of Shapley values, to address the challenge of fair benefit distribution in

collaborative environments. *Lazebnik, Beck & Shami (2023)* have utilized Shapley value method to address the issue of equitable benefit distribution among collaborating researchers, thereby identifying key nodes and influencers within collaboration networks. Similarly, *Akkas & Azad (2024)* used Shapley values with graph neural networks (GNNs) to develop a scalable and accurate framework for interpreting node importance. Their results further demonstrate the effectiveness of this approach in identifying core researchers within scientific collaboration networks (*Akkas & Azad, 2024*).

These methods, which integrate computational techniques with human cognitive capabilities, have improved the credibility and accuracy of identifying key research teams and analyzing the structure and dynamics of scientific collaboration. They provide valuable insights into the collaborative behaviors and influence patterns within scientific networks, contributing to more informed and effective management of research initiatives. However, existing methods often rely on either research performance metrics or network structural features in isolation, overlooking the need for integrated analysis. This limitation hampers their ability to fully capture the complex and dynamic interactions that occur within core sub-teams. To address this gap, this study proposes a novel approach that jointly incorporates research impact indicators (such as publication and citation metrics) and structural characteristics of collaboration networks. By integrating Shapley value analysis with sub-network h-index calculations, our method not only assesses the individual influence of researchers but also evaluates their collaborative interdependencies within the team. This dual-perspective framework enables a more nuanced and accurate identification of core research sub-teams, offering deeper insights into team dynamics and collaborative effectiveness.

## PROBLEM DEFINITION

Given a scientific collaboration network $G = (V, E)$ the set of nodes in the network is $V = v_1, v_2, \ldots, v_m$, representing a total of $m$ authors in the scientific collaboration network. $E = e_1, e_2, \ldots, e_l$ represents the edges in the network, which corresponds to $l$ collaboration relationships among the authors. If author $v_1$ and author $v_2$ have co-authored an article, there is an edge $e_{v_1, v_2}$ between the two authors.

For a scientific collaboration network $G$, let the set of all potential sub-teams is $\{T_1, T_2, \cdots, T_p\}$ indicating that there are possible sub-teams in the network. Each sub-team $T_i$ corresponds to a sub-network of $G$, hence a sub-team $T_i$ is also called a sub-network $T_i$. Let $h(\cdot)$ represent the importance evaluation function for sub-teams and various possible node combinations (in this article, the H-index is used). The core sub-team of size $k$ in the scientific collaboration network is defined as:

$$T^* = \underset{T_i \in \{T_1, T_2, \cdots, T_p\} \wedge |T_i| = k}{\arg\max} Score(h(\cdot), T_i) \cdot \tag{1}$$

Here, $Score(\cdot, \cdot)$ is a value assessment scoring function used to evaluate the importance of sub-teams in the scientific collaboration network. This article uses the Shapley value analysis method, which measures each participant's contribution to the game's outcome in

a cooperative game. In this context, each sub-team can be considered a game participant, and the overall team contribution can be regarded as the game's outcome. This article combines academic influence dimension information and network structure dimension information to measure the importance of sub-teams in the network. After determining the size $Score(\cdot, \cdot)$ of the core sub-team, the sub-team $T^*$ with the highest score based on Eq. (1) is regarded as the core sub-team of the scientific collaboration network.

## CORE SUBTEAM IDENTIFICATION METHOD BASED ON SHAPLEY VALUE ANALYSIS

In a scientific collaboration network, identifying the core sub-team requires evaluating the research output of the team. The research value of article authors or author teams can be reflected through the academic value of their articles. Generally, the academic value of articles is assessed based on citation counts (*Silver et al., 2017*). This article considers both the quantity and quality of articles published by the team and uses the H-index as an indicator to evaluate the research output value of the team.

The H-index of a sub-team is defined as follows: if a sub team has published $N$ articles, and there are $h$ articles that have each been cited at least $h$ times, while the remaining $N - h$ articles have been cited no more than h times (*Dehury & Sahoo, 2022*). The H-index can be used as a basis for evaluating the research output value of the entire research sub-team, as well as for each sub-team, providing an important basis for identifying the core sub-team (*Yuan et al., 2021*). However, using the H-index alone to find the core author sub-team does not reflect the collaboration relationships and the importance of individual authors within the research team.

The influence of sub-teams needs to take into account the degree of collaboration between individual authors and team members, as well as the value of the collaborative research output of individuals and team members. To efficiently and accurately identify the core sub-team in a scientific collaboration network, this article proposes a method combining Shapley value analysis with the Monte Carlo Tree Search algorithm, The overall workflow of this approach is illustrated in Fig. 1.

### Shapley value of sub-teams

The Shapley value is a mathematical method proposed in 1953 to solve multi-person cooperative game problems (*Shapley, 1953*). This method allocates benefits based on the marginal contributions of the participating individuals, ensuring fairness and effectiveness in the allocation (*Lipovetsky, 2021*; *Tian et al., 2022*). In this article, we use the Shapley value's contribution measurement criterion to assess the contribution of subteams composed of multiple authors to the entire scientific collaboration network.

Let there be $m$ nodes in the scientific collaboration network, represented by the node set $V = \{v_1, \cdots, v_i, \cdots, v_m\}$. The nodes in the sub-network corresponding to a sub-team can be represented as the set $\{v_1, \cdots, v_k\}$, while the other nodes belonging to $T \backslash T_i$ are represented as $\{v_{k+1}, \cdots, v_m\}$. That is, given a complex network with m nodes, the target sub-network with $k$ nodes can be obtained through Shapley value calculation. In this

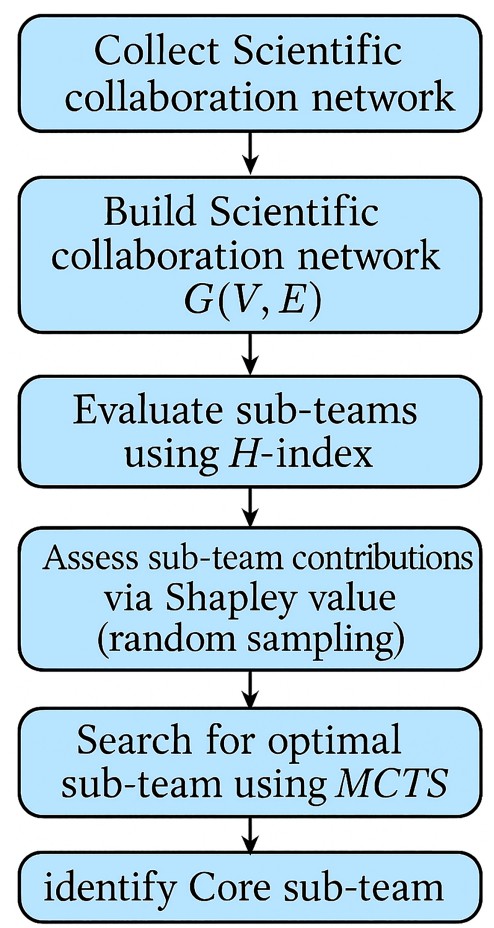

**Figure 1 Flowchart illustrating the methodology for identifying the core sub-team in scientific collaboration networks.**               

process, the set of authors participating in the Shapley value calculation is $P = \{T_i, v_{k+1}, \cdots, v_m\}$, where $T_i$ is a sub-network composed of multiple authors. The Shapley value of the sub-network $T_i$ can be defined as:

$$Shapley(T_i) = \frac{1}{|P|} \sum_{S \subseteq P \setminus \{T_i\}} \frac{1}{C_{|p|-1}^{|S|-1}} mc(S, T_i) \tag{2}$$

$$mc(S, T_i) = h(S \cup \{T_i\}) - h(S) \tag{3}$$

where $C$ represents the combination number, $mc(S, T_i)$ represents the marginal $S$ contribution of the given subnetwork $T_i$, obtained by comparing the difference in subteam value before and after combining the subnetwork $T_i$ and the set $S$. From Eq. (2), we can see that when calculating the Shapley value of a subteam, it considers the degree of impact on the entire team when that subteam is removed.

Based on Eqs. (2) and (3), we can obtain the calculation method for the Shapley value of subnetworks as shown in Algorithm 1. It is worth noting that the calculation of Shapley values involves traversing all possible combinations $P \setminus \{T_i\}$ in the set $S$, thus having a high

---

**Algorithm 1** Calculation of sub-team Shapley value.

**Input:** Scientific collaboration network $G$ with node set $V = \{V_1, \cdots, V_m\}$, sub-team $T_i$ containing $k$ nodes $\{V_1, \cdots, V_k\}$, and $n$ is the total number of sets $S$.

**Output:** Shapley value of the sub-team $T_i$.

**Initialization:** Obtain the neighboring nodes of sub-network $\{V_{k+1}, \cdots, V_m\}$. The set of authors participating in the calculation is $P = \{T_i, V_{k+1}, \cdots, V_m\}$.

1. **for** $i = 1$ **to** $n$ **do**

2.     Remove nodes and corresponding edges in $V \backslash (S_i \cup \{T_i\})$ from $G$.

3.     Calculate $h(S_i \cup \{T_i\})$.

4.     Remove nodes and corresponding edges in $V \backslash S_i$ from $G$.

5.     Calculate $h(S_i)$.

6.     Calculate this is the marginal contribution of the sub-network $T_i$ to the subset

$$S_i \; mc(S_i, \; T_i) = h(S_i \cup \{T_i\}) - h(S_i).$$

7. **end for**

**Return:** $Shapley(T_i) = \dfrac{1}{n} \sum\limits_{i=1}^{n} mc(S_i, \; T_i)$

---

complexity. In the following text, an approximate estimation method is adopted to improve computational efficiency.

## Core team's search and identification

In a scientific collaboration network, the size of the core sub-team is not fixed, often requiring a search for sub-teams of any size to determine the optimal core sub-team. Algorithm 1 targets sub-teams of a given size $k$, so it needs to be extended to handle variable sizes. This increases the computational complexity of the algorithm. To reduce computational costs, different sizes of sub-teams are structured into a tree and the MCTS algorithm is used for efficient searching. Additionally, during the Shapley value calculation of sub-networks, random sampling is used for approximate estimation, further improving computational efficiency.

MCTS is a best-first search algorithm, iteratively using random simulations to obtain effective search results within the tree. The MCTS algorithm can record the number of visits and the statistical reward data to guide exploration and reduce the search space. Each iteration of the search consists of four phases: selection, expansion, simulation, and backpropagation (*Teixeira da Silva & Dobránszki, 2018*). Generally, in MCTS, if a tree node represents a non-terminal state and has unvisited child nodes, then it is expandable (*Rossi, Winands & Butenweg, 2022*).

In the search tree constructed in this article, the root node corresponds to the scientific collaboration network, and the edges in the search tree represent that the corresponding sub-network of the child node can be obtained by partitioning the network corresponding to its parent node. We define $N_0$ as the root node in the search tree and $N_i$ as a node in the search tree. The edges in the search tree represent the partition step $p$.

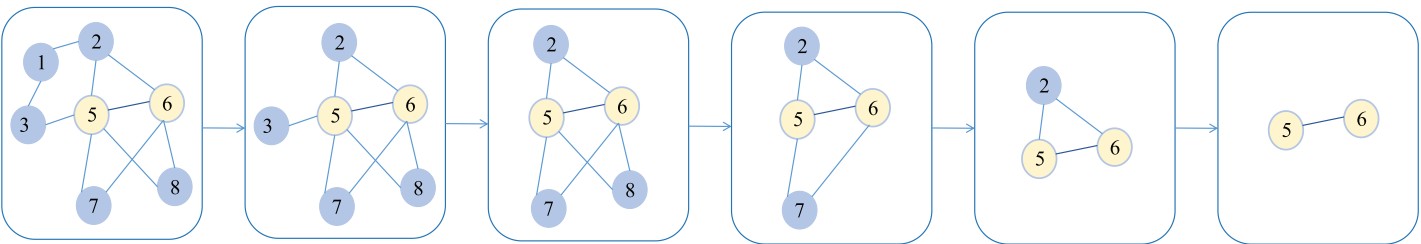

**Figure 2 The process of tree node selection in the MCTS algorithm for identifying core two-person seven sub-teams.**

Let the sub-network $T_j$ be obtained from $T_i$ by performing action $p_i$ denoted as $(N_i, p_j)$. To more clearly describe the MCTS algorithm, the following definitions are made:

- $\tau(N_i, p_j)$ represents the number of times the simulation operation $p_j$ has been performed on node $N_i$
- $Q(N_i, p_j)$ represents the total reward for all visits to $(N_i, p_j)$
- $V(N_i, p_j) = Q(N_i, p_j)/\tau(N_i, p_j)$ represents the average simulation reward.
- $R(N_i, p_j)$ represents the direct reward for performing simulation operation $p_j$ on node $N_i$. The direct reward value corresponds to the importance of the sub-network, and is defined as $R(N_i, p_j) = Score(h(\cdot), T_j)$.

Starting from the root node, the search strategy is depth-first through the tree until it can no longer expand, thus striking a balance between exploitation and exploration. The purpose of exploitation is to select nodes that can lead to the best results obtained so far, while the purpose of exploration is to explore nodes that exist due to evaluation uncertainty. The criteria for node selection are:

$$p^* = \arg\max_{p_j} V(N_i, p_j) + \lambda R(N_i, p_j) \frac{\sqrt{\sum_k \tau(N_i, p_k)}}{1 + \tau(N_i, p_j)} \tag{4}$$

where $\lambda$ is a hyperparameter that balances between exploration and exploitation, $N_i$ represents the total visit count of all possible actions for the node. The next step is to use a scoring function $Score(h(\cdot), T_j)$ to evaluate the importance of subnetworks, then proceed to the next iteration, propagating the simulation results backward, and updating all involved nodes and actions as follows:

$$\tau(N_i, p_j) = \tau(N_i, p_j) + 1 \tag{5}$$
$$Q(N_i, p_j) = Q(N_i, p_j) + Score(h(\cdot), T_j). \tag{6}$$

After multiple iterations, the subnetwork with the highest score can be calculated, and this highest-scoring subnetwork is considered the most core subteam. It is worth noting that in the early stages of the MCTS search process, there's a tendency to visit nodes with low visit counts to explore different partitioning operations. In later iterations, MCTS selects to visit nodes that produce higher rewards (*Rossi, Winands & Butenweg, 2022*). Figure 2 illustrates the tree node selection process of the MCTS algorithm when identifying core two-person subteams.

---

**Algorithm 2 Identify core sub-team.**

**Input:** H-index calculation program, MCTS iteration number **M**, minimum subnetwork node number $N_{min}$, the related subnetwork of tree node $N_i$ is $h(N_i)$. Scientific collaboration network $G$ with node set $V = V_1, \cdots, V_m$, the sub-network $T_j$ with $k$ nodes $V_1, \cdots, V_k$, random sampling number $r$.

**Output:** Core Sub-team $S_{core}$

**Initialization:** For each split operation $(N_i, p_j)$, initialize $\tau$, $Q$, $V$, $R$ to 0, The selected tree node set is $S_l =$, obtain the neighboring nodes of the subnetwork $V_{k+1}, \cdots, V_m$, the set of authors involved in the calculation is $P = T_i, V_{k+1}, \cdots, V_m$

1.    **for** $i = 1$ **to** $M$ **do**

2.       **if** $N_i$ has not been visited **then**

3.          **while** the number of nodes contained in $h(N_i)$ is greater than $N_{min}$ **do**

4.             for all possible split operations of $h(N_i)$ do

5.                Obtain the selected search tree node $N_j$ and subnetwork $T_j$x

6.                **for** $j = 1$ to r **do**

7.                    Sample a possible set $S_j$ from $P \backslash T_j$

8.                    Delete the edges between nodes in $V \backslash (S_j \cup \{T_j\})$

9.                    Calculate $h(S_j \cup \{T_j\})$

10.                  Delete the edges between nodes in $V \backslash S_j$

11.                  Calculate $h(S_j)$

12.                  $mc(S_j, T_j) = h(S_j \cup \{T_j\}) - h(S_j)$

13.               **end for**

14.               $Score(h(\cdot), T, T_j) = \dfrac{1}{r}\sum_{j=1}^{r} mc(S_j, T_j)$

15.            **end for**

16.            Select the next tree node $N_{next}$ according to Eq. (4)

18.         end if

19.    $S_l = S_l \cup \{N_i\}$

20.    Select the next tree node $N_{next}$ according to Eqs. (5) and (6)

21. **end for**

22. Select the core subteam $S_{core}$ with the highest score

---

Due to different nodes having different numbers of neighbors, there are still a large number of nodes in $P$, which affects computational efficiency. Therefore, this article further combines random sampling to approximate the calculation. During the sampling step $r$, subset $S_i$ is randomly drawn from $P$, and its marginal contribution $mc(S_i, T_i)$ is calculated. The average contribution score of multiple sampling steps $Shapley(T_i)$ is considered as an approximation, with the specific formula as follows:

$$Shapley(T_i) = \frac{1}{r}\sum_{i=1}^{r}(h(S_i \cup \{T_i\}) - h(S_i)) \tag{7}$$

where r is the total number of sampling iterations. In calculating the marginal contribution, this article adopts a zero-filling strategy. That is, when calculating $h(S_i \cup \{T_i\})$, we set the features of nodes not belonging to the subnetwork $V \backslash (S_i \cup \{T_i\})$

to zero and remove the edges connected to them. We then calculate the H-index $h(S_i \cup \{T_i\})$ of the team corresponding to the network $S_i \cup \{T_i\}$. Similarly, the value of $h(S_i)$ is the team's H-index after setting the node features to zero and removing the connected edges. Finally, we calculate the average of the differences obtained from multiple samplings, which is the Shapley value of each network. The subteam with the largest Shapley value is considered as the core subteam within the team (*Heskes et al., 2020*).

# EXPERIMENTATION

This section uses empirical analysis methods to collect real data on scientific collaboration networks, construct an effectiveness evaluation method, and analyze the effectiveness of the core team identification method based on Shapley value analysis in scientific collaboration networks.

## Experimental setup

### Datasets

To collect real data on scientific collaboration networks, we select researchers from a comprehensive university located in Shanghai that is known for its interdisciplinary strengths in the fields of industrial engineering, artificial intelligence, and automation. Then we gathered information on collaborating institutions and scholars from Baidu scholar homepages. Subsequently, under the same conditions of affiliation and research field, we identified scholars with co-authored publications using the Baidu Scholar and Web of Science (WoS) platforms. We recorded detailed information such as collaborative article content, citation count, journal impact factor, and retrieval information. We then repeated this process for the filtered scholars, collecting information on other collaborating scholars and articles (including both Chinese and English articles) in the same research field. Given the large number of scholars collected, we used Price's law: $M = 0.749 \left( \eta_{max} \right)^{1/2}$ to determine the final scientific collaboration team, where $M$ represents the minimum number of publications for core scholars, and $\eta_{max}$ is the number of articles by the author with the highest publication count. For example, in Team 1, the scholar with the highest publication count in the team had 131 articles, so we can determine $M = 8.573$. Therefore, we selected scholars with nine or more publications to form the scientific collaboration team.

Furthermore, during the data preprocessing stage, this study considered the issue of data disambiguation. First, regarding the disambiguation of domestic institutions, considering that inconsistencies in institution names are relatively rare, we manually checked cases where English names and pinyin names of institutions were inconsistent. We searched for the English and pinyin names of institutions in search engines and compared the results to confirm whether they were the same institution. The inconsistency in author names mainly occurred in the order of surname and given name in English literature. Within the same research institution, if the secondary units were the same, authors with the same surname and given name or given name and surname were considered as the same researcher. This study also concealed team names and author names, using team numbers to distinguish different teams and author numbers to

distinguish different authors within teams. We have incorporated all co-authorship relationships into the construction of the overall collaboration network. This includes both intra-team collaborations (within sub-team cores) and inter-team or external collaborations (with researchers outside the core team). These relationships are all represented as edges in the scientific collaboration network.

Analysis of the team size, number of collaborative articles, and average citation distribution of articles for the 61 teams shows that 34 teams have fewer than 10 members, 24 teams have 11 to 20 members, and three teams have more than 20 members. The number of collaborative articles per team ranges from a minimum of 15 to a maximum of 411, with 10 teams having more than 150 collaborative articles. The average citation count per article for each team ranges from a minimum of 1.9 to a maximum of 60.7, with 34 teams having an average citation count between 20 and 40 per article. For the sake of anonymization, we have concealed team names and author names, using team numbers to distinguish different teams and author numbers to distinguish different authors within teams. After applying this criterion, 61 collaboration teams were selected for the study.

### Evaluation of core team identification methods

In scientific collaboration networks, the core team usually consists of fewer members but produces the majority of the results. We evaluate the effectiveness of core team identification methods from two perspectives: the number of core team members and their contributions. Let $N$ be the number of nodes in the research collaboration network (*i.e.*, the total number of people in the research team), and $N_k$ be the number of core team members, then $P_m = N_k/N$ represents the proportion of core team members. Let $O$ be the total contribution of the research team, and $O_k$ be the contribution of the core team, then $P_c = O_k/O$ represents the proportion of the core team's contribution. Given a scientific collaboration network, as the number of nodes judged to be core members increases, the proportion of the core team's contribution also increases. Therefore, $P_m$ and $P_c$ will form a curve in the coordinate axis from point (0, 0) to point (1, 1), which we term the Member-Contribution (MC) curve.

Given a scientific collaboration network, if the MC curve of one core team identification method envelops the MC curve of another method, then the former has a more accurate identification effect. As shown in Fig. 2, the method represented by $MC_2$ is superior to the method represented by $MC_1$. To make a more accurate quantitative comparison of identification methods, this article uses the area under curve (AUC) of the MC curve, called MC-AUC, to compare the effectiveness of different methods in identifying core teams in scientific collaboration networks. As shown in Fig. 3, the shaded area represents the area under the curve of $MC_1$, denoted as MC-AUC.

A problem that exists in the calculation of the above indicators is the method of quantifying the core team's contribution. The contribution of a research team includes various factors such as different types of scientific output, talent training, *etc.*, which is an important issue in scientific research evaluation. For the sake of simplicity, this article uses the number of published articles to represent the team's collaborative contribution, and the

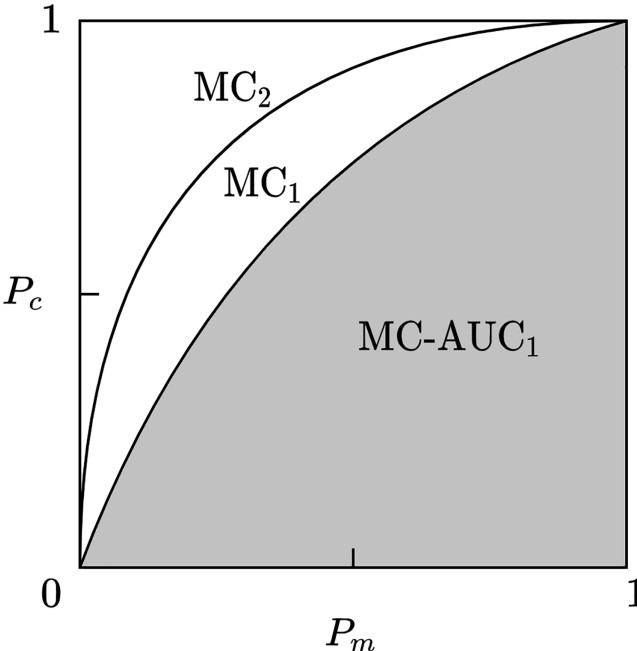

**Figure 3  Member-contribution curves and area comparison for core team identification methods.** The MC1 and MC2 curves represent the cumulative contributions of the identified core team members. The area under each curve, referred to as MC-AUC, is used as a quantitative metric to evaluate and compare the performance of different core team identification methods.

number of citations of articles as the team's impact contribution. The effectiveness of core team identification methods is analyzed from these two perspectives respectively.

In this study, nodes are first ranked based on the evaluation metrics for identifying core teams in the scientific collaboration network. Then, different sizes of node sets are sequentially selected as core teams.

(1) The degree of a network node is the number of its edges, representing the collaboration relationships between researchers in a scientific collaboration network.

(2) Node strength is the sum of the edge weights of a node, with the edge weight represented by the number of collaborations (*i.e.*, the number of jointly published articles).

(3) The k-core is a subnetwork composed of nodes with a minimum degree of k. In network analysis, degree and node strength focus only on the number of connections of a node, making them local metrics used to evaluate individual nodes. The k-core, on the other hand, considers the extent to which a node belongs to the core structure of the entire network, focusing on nodes in the largest connected subgraph, making it a global metric.

(4) Betweenness centrality describes the extent to which a node acts as an intermediary in information transfer within the network, helping us understand which subteams play important roles in information dissemination within the scientific collaboration network.

(5) Closeness centrality measures the average shortest path length between a node and all other nodes, helping us understand which nodes are more closely connected to other nodes within the network.

**Table 1 Core team identification results.** The bolded entries achieve the best performance.

| Types | Group size | Shapley | Degree | Strength | Kcore | Betweenness | Closeness |
|---|---|---|---|---|---|---|---|
| Collaborative | <7 (22) | **0.818** | 0.136 | 0.136 | 0 | 0.136 | 0.136 |
| | 7–10 (18) | **0.667** | 0.222 | 0.222 | 0 | 0.222 | 0.278 |
| | >10 (21) | **0.524** | 0.095 | 0 | 0 | 0.429 | 0.19 |
| | Total (61) | **0.672** | 0.148 | 0.115 | 0 | 0.262 | 0.197 |
| Impact | <7 (18) | **0.727** | 0.136 | 0.227 | 0 | 0.136 | 0.136 |
| | 7–10 (18) | **0.722** | 0.222 | 0.111 | 0 | 0.278 | 0.222 |
| | >10 (21) | **0.619** | 0.143 | 0 | 0 | 0.286 | 0.19 |
| | Total (61) | **0.689** | 0.164 | 0.115 | 0 | 0.23 | 0.18 |

## Results

The number of members in a scientific collaboration network varies greatly. In the data collected for this study, the numbers range from a few to several dozen. This variability can affect the results of core team identification. To minimize this impact, the data is divided into three groups based on the number of nodes: less than 7 (22 networks), 7 to 10 (18 networks), and more than 10 (21 networks). Within each group, the core teams identified by different methods are ranked by their contribution ratios. The higher the ratio, the more accurate the identification results. As shown in Table 1, the proposed method achieved the best results in all groups. Each value in the table represents the proportion of times a method ranked first within its respective group. For example, the Shapley value analysis method achieved the most accurate identification results in 81.8% of the 22 teams with fewer than seven members.

It is noteworthy that the k-core decomposition method did not achieve the best identification results in any analysis. This is because k-core decomposition is a relatively coarse-grained network node ranking method with very low MC curve smoothness, resulting in a lower score when calculating MC-AUC. From the experimental results, it is evident that betweenness centrality and closeness centrality metrics performed second best to the Shapley value metric and were significantly better than degree, node strength, and k-core. However, when complex node relationships and connection patterns exist in the network, betweenness centrality and closeness centrality metrics have their limitations. They do not consider the role and contribution of nodes when combined with other nodes.

In contrast, the Shapley value considers not only the direct connections and interactions within subteams but also the role and contribution of subteams when combined with different teams. This allows for a more comprehensive description of the importance of subteams.

As shown in Fig. 4, using collaboration and impact as contribution metrics, the results obtained by different methods in the MC-AUC calculation are quite similar. This is partly because there is a high correlation between the number of publications and the number of citations. Meanwhile, the consistent performance across both metrics also demonstrates the stability and reliability of MC-AUC as an evaluation indicator for core team identification. Furthermore, one-way ANOVA tests conducted on the experimental results

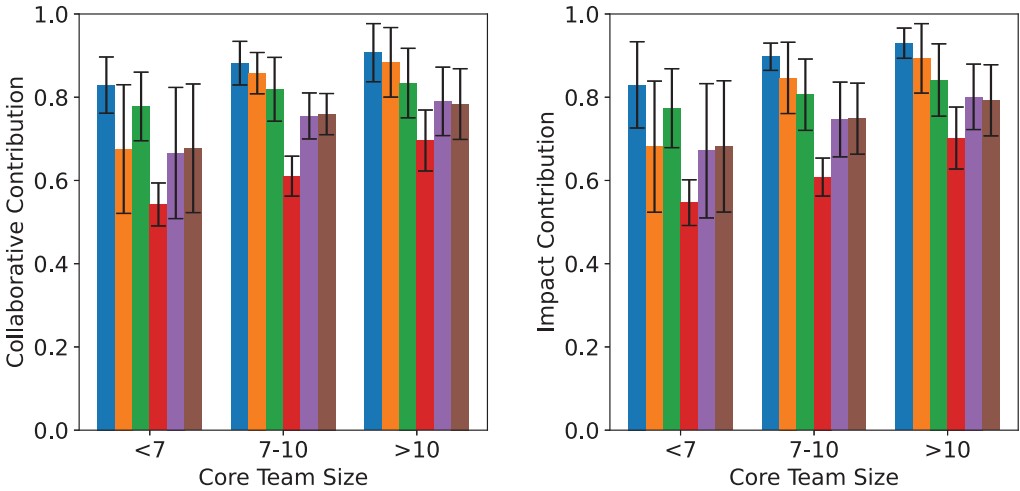

**Figure 4 Comparative analysis of different core teams selection methods across different sizes.** The left and right panels display the average collaborative and impact contributions of core teams in three size groups (<7, 7–10, >10), based on six identification methods. Error bars indicate standard deviations across datasets.

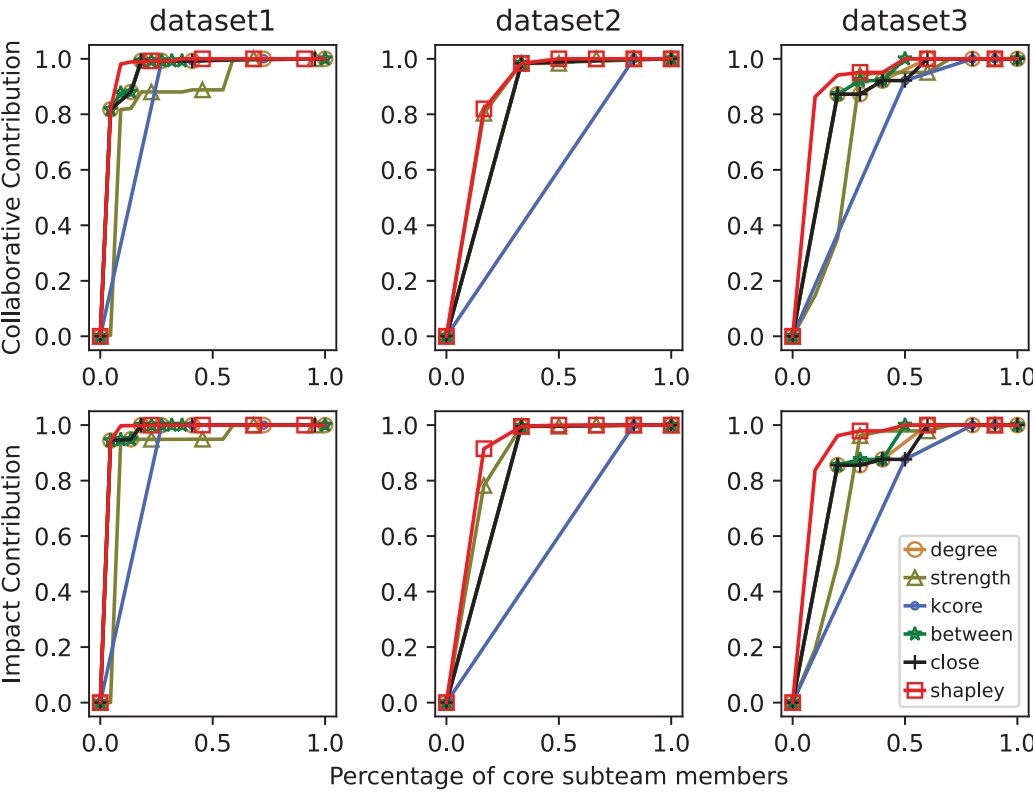

**Figure 5 MC curves of core sub-team identification methods across datasets.** The top row shows collaborative contributions, while the bottom row shows impact contributions. The x-axis represents the proportion of selected core members, and the y-axis reflects cumulative contribution. A higher curve indicates better identification performance.

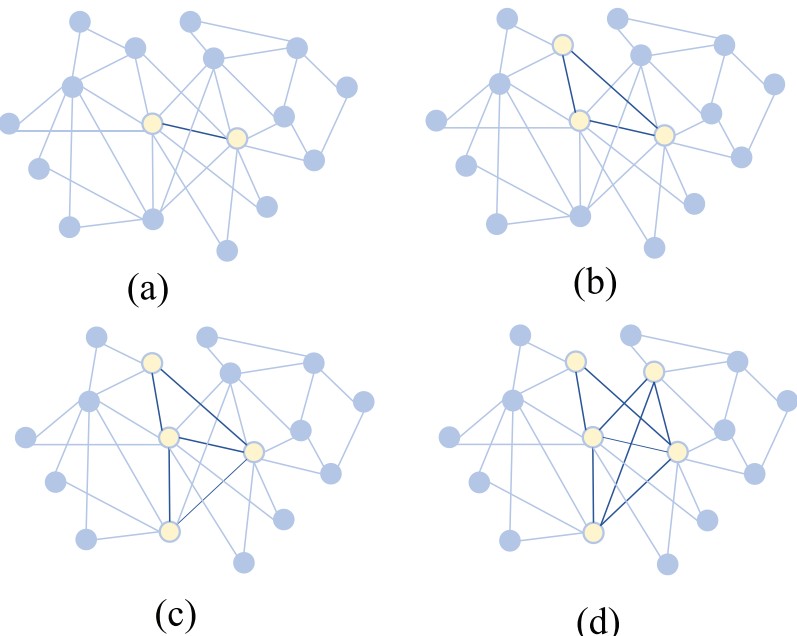

**Figure 6 Identification of core sub-teams with different sizes in real-world scientific collaboration networks using Shapley value and MCTS.** (A) 2-member core (B) 3-member core (C) 4-member core (D) 5-member core. Yellow nodes represent selected core members, and dark edges indicate stronger connections.

confirm that the differences in contribution scores across methods are statistically significant (collaborative contribution: F = 4.91, $p$ = 0.011; impact contribution: F = 4.94, $p$ = 0.011), thereby reinforcing the conclusion that our proposed method exhibits superior performance in a statistically meaningful manner.

In Fig. 5, MC curves of core sub-team identification methods across datasets. Since the problem addressed in this article involves the extent of information transmission within the network and its relation to collaborative publications, as well as the distance and connection degree between nodes related to collaborative articles, there is a certain degree of correlation between the two. Therefore, in smaller scientific networks, the results identified by betweenness centrality and closeness centrality metrics are similar.

## Case analysis

To intuitively validate the effectiveness of the proposed method, this section constructs a scientific collaboration network based on a real-world research team, illustrating the identification of core sub-teams using the Shapley value and MCTS algorithm.

Specifically, the scientific collaboration network consists of 15 authors with a total of 157 co-authored articles. As shown in Fig. 6, the core subteams identified by the algorithm with sizes of 2, 3, 4, and 5 are displayed within this network. Experimental results indicate that the most important 5-member sub-team includes Authors 1, 2, 3, 4, and 5. Among them, Authors 1, 2, 4, and 5 are from the same research institution, while Author 3,

**Table 2 H-index of authors in the research team.**

| Author in the team | Individual H-index | Author in the team | Individual H-index |
|---|---|---|---|
| Author 1 | 12 | Author 9 | 10 |
| Author 2 | 12 | Author 10 | 9 |
| Author 3 | 32 | Author 11 | 4 |
| Author 4 | 44 | Author 12 | 3 |
| Author 5 | 7 | Author 13 | 5 |
| Author 6 | 12 | Author 14 | 8 |
| Author 7 | 24 | Author 15 | 9 |
| Author 8 | 5 | | |

although from a different institution, collaborates closely with Authors 1, 2, and 4 within this five-member sub-team. Author 3 has contributed 27 articles to the team, with a notably high citation count.

When considering smaller core sub-teams, the algorithm identifies Authors 1, 2, 3, and 4 as the most important four-member group. For the three-member sub-team, the key members are Authors 1, 3, and 4, while for the two-member sub-team, the most important pair is Authors 3 and 4. Notably, when we set the sub-team size to 1, we can identify the most important individual authors within the collaboration network. It is observed that authors with high publication counts and citation rates, such as Authors 3 and 4, are consistently included in key sub-teams.

The Shapley values calculated differ for sub-teams of different sizes. Generally, the more members a subteam has, the higher its Shapley value. However, an interesting observation in Team 2 reveals that the four-member sub-team of Authors 1, 2, 3, and 4 has a Shapley value of 10.7, while the five-person subteam consisting of Authors 3, 4, 9, 10, and 14 has a Shapley value of 8.47. This suggests that a smaller sub-team can sometimes hold greater importance than a larger one, highlighting that quality of collaboration can outweigh sheer team size. In addition, analysis shows that the four-person subteam of Authors 1, 2, 3, and 4 has authored 71 articles within the team and has an H-index of 19. In contrast, although the five-person subteam of Authors 3, 4, 9, 10, and 14 contributed 81 articles to the entire research team, this subteam's H-index is only 14.

This finding demonstrates that the Shapley value is not directly affected by the number of sub-team members. Instead, it is influenced by collaborative quality, as indicated by the H-index and publication contributions.

An examination of individual H-indices further illustrates that a high H-index does not necessarily correlate with greater team importance. As shown in Table 2, author 7 has an H-index of 24 in Table 2, but is not included in the calculated most important subteam. Author 6 has an H-index of 12, equal to Authors 1 and 2, but is also not included in the most important subteam. Analysis reveals that Author 7 only contributed three publications to Research Team 2 and only collaborated with Authors 12 and 13. Author 6 collaborated with Authors 2, 3, and 4, but only on eight articles with relatively few

**Table 3 Difference in Shapley values after removing authors from sub-teams.**

| Removed author | 5-person team | | 6-person team | | 7-person team | | 8-person team | |
|---|---|---|---|---|---|---|---|---|
| | Difference in Shapley value | Contribution ratio of Shapley | Difference in Shapley value | Contribution ratio of Shapley | Difference in Shapley value | Contribution ratio of Shapley | Difference in Shapley value | Contribution ratio of Shapley |
| Author 1 | 3.04 | 13.58 | 2.4 | 11.90 | 2.39 | 11.56 | 2.13 | 11.64 |
| Author 2 | 1.44 | 4.65 | 2.36 | 8.50 | 2.31 | 7.53 | 1.78 | 7.61 |
| Author 3 | 3.87 | 28.47 | 3.66 | 27.13 | 3.63 | 26.34 | 3.96 | 25.99 |
| Author 4 | 5.35 | 44.93 | 3.87 | 42.09 | 3.96 | 40.32 | 4.58 | 40.30 |
| Author 5 | 1.25 | 8.37 | 1.88 | 7.40 | 1.33 | 6.72 | 1.91 | 6.81 |

citations. Thus, we can conclude that Shapley values are influenced not only by H-index but also by collaborative relationships between authors in the team.

To analyze individual impacts within sub-teams, Shapley values were recalculated after removing specific authors, with differences recorded to measure influence. Calculating Shapley value differences is a common sensitivity analysis method. Removing a node in a network may affect how other nodes connect, thus impacting their Shapley values. Computing Shapley value differences can further investigate an author's importance in a subteam and their degree of influence. A larger difference after removing an author indicates a greater influence on the subteam. We also calculated individual Shapley value contribution ratios by dividing an individual's Shapley value by their subteam's Shapley value. This ratio reflects each feature's relative importance to the model's prediction results, decomposing the model's predictions into individual author contributions and showing each author's impact on the prediction results. This helps us better understand how the model makes predictions and explain its decision-making process.

According to Table 3, vertical comparisons show that removing Author 4 results in a significantly larger Shapley value difference compared to removing other authors, indicating Author 4 has a greater influence in the subteam. Removing Author 5 results in a smaller Shapley value difference, suggesting Author 5 has a relatively smaller influence. Author 4 has the largest Shapley value contribution ratio, indicating their contribution has the greatest impact on prediction results. Horizontal comparisons reveal that each author's degree of influence varies in different subteams. The Shapley value contribution ratio is not affected by team size but changes based on the collaboration situation of team members.

## CONCLUSION

This study investigates the identification of core sub-teams in scientific collaboration networks by integrating Shapley value analysis and MCTS algorithm. By leveraging the H-index to reflect research impact and utilizing MCTS to efficiently explore sub-team combinations, the proposed approach evaluates each sub-team's importance based on both academic productivity and collaborative structure. Empirical results demonstrate that this method effectively identifies core sub-teams in research collaboration networks. Compared

to traditional team evaluation metrics, our approach yields more stable and interpretable results across different team sizes and domains.

The findings show that the sub-teams identified using the Shapley value and MCTS-based method contribute significantly to the achievements of the entire team, often assuming more critical roles within the team structure. The case analysis further supports these results, demonstrating that these core sub-teams exhibit closer relationships and more stable collaborations among their members. This method, therefore, holds theoretical promise for application across various types of research collaboration networks, offering a nuanced evaluation of team contributions.

Based on our findings, we offer several recommendations to guide future research and practical applications. First, institutions and funding agencies should incorporate external collaboration indicators—such as cross-team co-authorship and structural hole positions—to better capture a team's integrative influence. Second, adopting longitudinal analysis tools can help track the evolution of research collaborations and identify emerging core sub-teams over time. Third, evaluation frameworks should combine bibliometric indicators with network-based metrics to enable more comprehensive assessments of researcher contributions. Finally, analytics platforms would benefit from integrating computational optimization techniques like MCTS to efficiently identify high-impact sub-teams within large-scale collaboration networks.

However, there are limitations to this study. In real-world research settings, individuals often engage in cross-team or inter-institutional collaborations, which serve as important channels for acquiring new knowledge and techniques. Future work could enrich our framework by incorporating measures of external collaboration embeddedness or dynamic knowledge transfer Additionally, the current analysis is based on static networks. Extending the method to dynamic collaboration networks would enable tracking the evolution of core teams over time and provide deeper insight into their development trajectories.

### Funding
This work was supported by the National Social Science Foundation of China (No. 22CGL050). The funders had no role in study design, data collection and analysis, decision to publish, or preparation of the manuscript.

### Grant Disclosures
The following grant information was disclosed by the authors:
National Social Science Foundation of China: 22CGL050.

### Competing Interests
The authors declare that they have no competing interests.

## Author Contributions

- Lixin Zhou conceived and designed the experiments, performed the experiments, analyzed the data, performed the computation work, prepared figures and/or tables, and approved the final draft.
- Chen Liu conceived and designed the experiments, performed the experiments, performed the computation work, authored or reviewed drafts of the article, and approved the final draft.
- Xue Song performed the experiments, analyzed the data, performed the computation work, prepared figures and/or tables, authored or reviewed drafts of the article, and approved the final draft.

## Data Availability

The data is available at Zenodo: Song, Xue (2025). core sub-team [Data set]. Zenodo. https://doi.org/10.5281/zenodo.14930241.

## Supplemental Information

Supplemental information for this article can be found online at http://dx.doi.org/10.7717/peerj-cs.3048#supplemental-information.

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
