# Peer review of "Identification of core sub-team on scientific collaboration networks with Shapley method"

_PeerJ Computer Science, doi:10.7717/peerj-cs.3048_

## Round 0.1 · original submission · Major Revisions

· Academic Editor

Major Revisions

Dear authors,

Thank you for submitting your manuscript "Identification of core sub-team on scientific collaboration networks with Shapley method" to PeerJ Computer Science. Based on the reviewers’ assessments, we are inviting a major revision. While both reviewers recognise the relevance and potential of your work, particularly the application of Shapley values to collaboration networks, they also identify significant issues that need to be addressed before the manuscript can be considered for publication. These include improving the clarity and structure of the manuscript, providing more detail on the dataset and methodological choices, ensuring the reproducibility of your analysis, incorporating stronger comparative validation, and refining the presentation of the problem, objectives, and conclusions. We encourage you to revise the manuscript accordingly and to submit a detailed point-by-point response to the reviewers' comments.

Best regards,

José M. Galán

Reviewer 1 ·

Basic reporting

This manuscript presents a novel framework for studying collaboration networks, which play a crucial role in understanding how social ties among researchers influence the outcomes of collaborative projects. The application of Shapley values, while established in the literature, is still relatively uncommon in this context, adding an interesting and valuable dimension to the study. The topic aligns well with the journal’s scope and has the potential to contribute meaningfully to the field.

That said, the manuscript appears to be in an early stage of development and would benefit from substantial improvements in terms of clarity, structure, methodological rigor, and presentation. Addressing these areas will be essential for realizing the full potential of the proposed approach.

Experimental design

The authors rightly note the growing availability of research data through platforms such as Web of Science (WoS). Given this, it is unclear why the final dataset includes only 61 collaboration teams. Could the authors clarify the selection criteria used to narrow down the dataset? Providing more detail on this point would help readers better understand the scope and limitations of the analysis.

It would also be helpful to include supplementary materials outlining the institutions involved in the study, as well as demographic information about the selected authors. This information is crucial for assessing the replicability—and even the reproducibility—of the findings. If such details are already included in the supplementary files, and I may have overlooked them, I would appreciate being directed to them.

I have also checked the data available in the Zenodo repository, and with that data, it is not possible to fully reproduce the findings, given that they are already processed and without any documentation.

The definition of a team's h-index could benefit from greater clarity. Given the somewhat ambiguous structure in the manuscript, where both teams and sub-team cores are mentioned, it’s not entirely clear how the h-index is calculated. Does it account only for papers co-authored by the same set of team members, or are contributions from external collaborators also included? This distinction is important, especially considering the potential impact of new researchers bringing additional expertise or resources to the team.

Another potential limitation of the current approach is that it may overlook individual collaboration networks. Researchers often engage in collaborations outside their core teams, which can be valuable for acquiring new knowledge or techniques that they later bring back to their main groups. Incorporating this dynamic could enrich the analysis and better reflect real-world collaboration patterns.

The analysis includes variables such as the number of publications, collaborators, and h-index, which are all likely to be highly correlated. This raises concerns about multicollinearity and its potential impact on the robustness of the results. Addressing this—either through statistical tests or by clarifying how these correlations are handled—would strengthen the credibility of the findings.

Validity of the findings

The authors suggest that their method offers advantages over other network science–based approaches for identifying core subteams, such as clustering techniques. However, a direct benchmark comparison with these established methods is missing. While Figure 3 includes comparisons with metrics like degree and k-core, these are not typically considered strong competitors for subgraph identification. Including a comparison with more sophisticated approaches, such as modularity-based clustering, community detection, or graph partitioning, would provide a clearer picture of the method’s relative strengths and limitations.

Additionally, from Figure 3, it is not possible to assess whether the observed differences in the score metric are statistically significant. Including some form of statistical analysis (e.g., hypothesis testing, confidence intervals, or effect sizes) would help validate the performance claims and support the conclusions drawn from the comparison.

Additional comments

Overall, I find the core idea of this work relevant. However, in its current form, there are several important concerns that remain unaddressed, which prevent me from recommending it for publication at this stage. I believe that with substantial revisions and clarifications, this manuscript could be significantly strengthened. I would be happy to review a severely improved version should the authors choose to address the points raised.

Annotated reviews are not available for download in order to protect the identity of reviewers who chose to remain anonymous.

Reviewer 2 ·

Basic reporting

The abstract is unclear with the problem statement and recommendations are not mentioned. The introduction section is okay but it may be written in more simpler words to identify the research gap, problem statement, significance and author’s contribution clearly.
The paper does not provide a comparison with the previous work.
Methodology and experimentation should be clearly defined in this section.
Aligned the Methodology section with results
Clearly write the objectives of the study
Results is good and well-written
Section 5 Empirical Analysis have subheading 4.1 Experimental Setup replace it with 5.1
All section/heading numbering should be revised
The conclusion section does not give the future work.
Also add recommendations after conclusion
The latest references from the 2025 have not been added. Add 1 or 2 references the year.
So, revise the paper with the above reason.

Experimental design

Add flow diagram for better understanding

Validity of the findings

Results should be imroved and provide analysis also
Add comparative analysis for validation

---

## Round 0.2 · accepted · Accept

· Academic Editor

Accept

Thank you for submitting the revised version of your manuscript entitled "Identification of Core Sub-team on Scientific Collaboration Networks with Shapley Method" to PeerJ Computer Science. We appreciate the time and effort you and your co-authors have dedicated to addressing the reviewers' comments.
After careful evaluation of your detailed responses and the revised manuscript, I am pleased to inform you that your submission is now acceptable for publication.


The reviewers raised important concerns regarding the clarity of your methodological framework, the replicability of your findings, and the presentation quality. Your rebuttal addresses these issues comprehensively. You have clarified the selection criteria for your dataset, improved the documentation and transparency of your data and code repository, and refined key definitions, including that of the team h-index.

Notably, the revised manuscript exhibits improved structure, clearer exposition, and strengthened statistical validation of your proposed method. These enhancements significantly elevate the contribution of your work to the study of scientific collaboration networks.

Congratulations on your revision. We look forward to publishing your work.
Sincerely,